# Stromal Hedgehog Signaling Is Associated with Favorable Outcomes in Pancreatic Cancer

**DOI:** 10.3390/ijms26115200

**Published:** 2025-05-28

**Authors:** Paul Manoukian, Helene Damhofer, Lan Zhao, Hanneke W. M. van Laarhoven, Maarten F. Bijlsma

**Affiliations:** 1Laboratory of Experimental Oncology and Radiobiology, Cancer Center Amsterdam, Amsterdam UMC and University of Amsterdam, 1081 BT Amsterdam, The Netherlands; p.manoukian@amsterdamumc.nl (P.M.); hd@biorigin.dk (H.D.); lanzhao20140101@gmail.com (L.Z.); 2Cancer Center Amsterdam, Cancer Biology, 1081 BT Amsterdam, The Netherlands; h.vanlaarhoven@amsterdamumc.nl; 3BiOrigin, 2200 Copenhagen, Denmark; 4Department of Medicine, Stanford University School of Medicine, Palo Alto, CA 94305, USA; 5Department of Medical Oncology, Amsterdam UMC and University of Amsterdam, 1081 HV Amsterdam, The Netherlands

**Keywords:** PDAC, hedgehog signaling, cancer-associated fibroblasts, tumor-stroma interaction

## Abstract

Aberrant activation of the Hedgehog (Hh) signaling pathway can be observed in various malignancies, particularly in stroma-rich tumors like pancreatic ductal adenocarcinoma (PDAC). In PDAC, Hh signaling is thought to foster an abundant stroma, making it an appealing target for stoma-targeted therapy. However, the use of Hh antagonists in the clinic has thus far not been successful. To reassess the clinical merit of Hh-targeted therapy in PDAC, we sought to better characterize the role of Hh signaling in tumor-stroma crosstalk. Here, we show that Hh ligands are not prognostic per se in PDAC, despite being associated with the favorable classical molecular subtype. Perturbing Hh ligand expression in PDAC cells can effectively alter their trans-signaling capacity but does not impact tumor growth in vivo. However, co-injecting PDAC cells with Smo-proficient MEFs resulted in a significant reduction in xenograft growth, suggesting that Hh-related effects on tumor growth are largely mediated through the stroma. By analyzing transcriptomic sequencing data from co-cultures, comprising human PDAC cells and mouse fibroblasts treated with a Hh-blocking antibody, we could identify stromal hits that are responsive to Hh ligands. We then leveraged the obtained set of genes to allow patient stratification based on stromal response to Hh ligands. We believe that a subset of PDAC patients may benefit from the use of Hh-targeted therapies and thereby encourage the use of our stratification tool to guide their use in PDAC clinical care.

## 1. Introduction

Pancreatic ductal adenocarcinoma (PDAC) is an aggressive malignancy that comprises roughly 90% of all pancreatic cancer cases [1]. Despite advances in therapies over the past decades, there have been only limited improvements in the survival outcomes of patients [2]. A main characteristic of this disease is the presence of a highly desmoplastic stroma that is composed of both cellular and non-cellular components [3]. PDAC stroma harbors various cell types (e.g., endothelial cells, pericytes, adipocytes, and immune cells) and can exhibit both tumor-promoting and -suppressing activities [4]. PDAC stroma contributes most to the abundance of fibroblast populations that are involved in the deposition of the extracellular matrix [4]. Considering the contributions of cancer-associated fibroblasts (CAFs) to pancreatic tumorigenesis and PDAC progression, many previous efforts have aimed to investigate and characterize these cells in hopes of guiding clinical practice. CAFs are heterogeneous in terms of phenotypic and morphological characteristics as well as functional properties such as (anti)tumorigenic capacity [4]. A myriad of markers have been described that delineate stromal heterogeneity within PDAC tumors [5,6,7]. Two main subsets of CAFs (myofibroblast-like CAFs and inflammatory CAFs) are now accepted to exist, with the former encompassing the major subpopulations [3]. Myofibroblast-like CAFs (myCAFs) lie proximal to the tumor cells and are characterized by their contractile capacity and high α-smooth muscle actin (αSMA). Inflammatory CAFs (iCAFs) are more distal to the tumor cells and are distinguished by their secretion of IL-6, among other inflammatory cytokines [8]. Additional novel subgroups of CAFs partly overlap with the aforementioned subsets [9]. Additionally, CAFs seem to be less heterogeneous compared to PDAC cells, suggesting the presence of common mechanisms for CAF activation and/or maintenance. Numerous signaling pathways are active in PDAC and together contribute to clinical outcomes [10]. Many of these pathways play a role in shaping the TME and are crucial in the recruitment and activation of CAFs. A prominent example of this is the TGF-β signaling pathway and the CXCL12-CXCR4 axis thought to contribute to the immunosuppressive and immune-excluded microenvironment of PDAC [11,12]. However, some signaling pathways have been described to exert tumor-suppressive functions; this has been, for the most part, attributed to the immune system [11]. For instance, type I interferons can induce various anti-tumor effects, both directly or through immunologic and vascular mediators [13]. IFN-γ can activate the CXCL9,10/CXCR3 axis, which is associated with a favorable prognosis in PDAC [14]. Moreover, macrophage-derived CXCL9 and CXCL10 are crucial for anti-tumor immunity following immune checkpoint blockade [15]. The contributions of CAFs to immune suppression in PDAC are not well understood. Moreover, there are indications that certain subsets of CAFs support anti-tumor immunity [5], urging a reassessment of how we target the stroma.

The Hedgehog (Hh) pathway has numerous key roles during embryogenesis [16]. It is involved in various tissue patterning and cell fate determination events [17]. Hh signaling is notably absent during the development of the pancreas, and activation of this signaling pathway is thought to repress pancreatic development [18,19]. However, aberrant activation of Hh signaling can be observed in adults, typically associated with malignancies [16]. This is evident in stroma-rich tumors like PDAC, where Hh signaling fosters an abundant stroma [20]. While early preclinical studies have suggested Hh targeting to suppress PDAC growth and metastasis, subsequent clinical trials have shown that antagonists of this pathway can actually accelerate PDAC progression [20]. It is currently unclear whether Hh targeting could be an effective therapeutic strategy for PDAC. Canonical hedgehog signaling is initiated by the binding of Hh ligands (Sonic Hedgehog, SHH; Indian Hedgehog, IHH) to Patched (PTCH1, PTCH2), thereby inhibiting Patched, which then relieves Smoothened (SMO) inhibition. In turn, SMO activates downstream signaling, enabling the translocation of activation of Gli transcription factors (GLI1, GLI2) to the nucleus, wherein they regulate the expression of target genes [21]. Hh cross-talk can occur in both an autocrine or paracrine manner; however, secretion and consequent paracrine signaling are contingent on the expression of dispatched (DISP), which is required for the release of the lipophilic Hh ligands [22].

In PDAC, tumor cells secrete Hh ligand, while surrounding CAFs respond to these ligands and are subsequently activated [23]. Hh pathway activation impacts CAF states; inhibiting this pathway in a PDAC mouse model using the smoothened antagonist LDE225 was shown to reduce myCAF numbers while simultaneously enriching iCAFs, indicating a role of Hh signaling in shaping CAF phenotypes [24]. Other recent studies have shed light on the complex role of Hh pathway inhibition in PDAC and may explain the observed negative clinical outcomes. While Hh-targeted therapies have an anti-fibrotic role and reduce myCAFs (Gli1+ stromal cells), they simultaneously encourage aggressive mesenchymal tumor cell features [25]. A similar observation was noted in mouse models of PDAC that were either genetically depleted for Shh or subjected to Smo inhibition and revealed a tumor-suppressive role for CAFs through restraining tumor angiogenesis [25]. This is concordant with the finding that αSMA+ myofibroblast depletion in PDAC can result in enhanced EMT and cancer stem cell features, alongside a decrease in overall immune infiltration and increased frequency of FoxP3+ regulatory T-cells [26]. Also, inhibition of Hh pathway activity by vismodegib leads to a significantly higher proportion of PanIN lesions within pancreata, suggesting an inhibitory role for the Hh pathway during early pancreatic tumorigenesis [20]. In fact, early Hh pathway activation in KPC mice exhibits a greater effect on stroma remodeling compared to a murine PDAC transplantation model, suggesting a crucial role in early pancreatic tumorigenesis [24]. In the setting of established PDAC, the Hh ligand SHH is inversely correlated with Gli activity in cancer cells [27]. Moreover, Hh pathway activation dramatically reduces the expression of Pdx1, suggesting that the progenitor-like phenotype is inversely correlated with Hh pathway activation and that Hh inhibition may result in poorly differentiated PDAC phenotypes [20]. PDAC cancer cells also vary in their expression of Gli transcription factors, and Gli2 in particular has been shown to associate with poor prognostic mesenchymal subtype PDAC [28]. It is thus crucial to understand the role of Hh during early tumorigenesis and PDAC progression in order to effectively implement stroma-targeted therapies in the clinic. To that end, we sought to re-evaluate the contributions of Hh signaling to PDAC with the aim to guide potential future stromal interventions.

## 2. Results

### 2.1. Hedgehog Ligand Abundance Is Not Prognostic in PDAC

Previous literature as well as work by our group has shown that PDAC cells secrete elevated levels of Hh ligands (SHH, IHH), which are thought to confer tumor-promoting stromal features. Fibroblasts respond to these ligands and upregulate target genes GLI1 and PTCH1. We first confirmed this by comparing the expression of HH ligands and known downstream signaling elements using a previously described patient-derived xenograft (PDX) RNA sequencing dataset [27]. This enabled us to discern between human and mouse reads and ascertain whether the transcripts were of tumor or stromal source, respectively (Appendix A). We observed that Hh ligands SHH and IHH (not DHH) were expressed by PDAC cells, whereas downstream genes had a higher expression in the stromal compartment. Considering the key role of Hh signaling in PDAC and its influence on both stromal and cancer cell features, we sought to determine whether Hh ligand abundance could impact the aggressiveness of PDAC cells and thereby be of clinical significance. To begin with, we used public RNA-seq data from the Cancer Cell Line Encyclopedia (CCLE) to confirm the correlation between Hh ligands and PDAC transcriptomic subtypes. The transcriptomic subtypes of PDAC hold different prognoses and have been shown to represent different cellular states of PDAC that lie on opposing ends of the EMT axis. This showed a negative and positive correlation, respectively, with signatures for basal and classical PDAC (Figure 1A). To corroborate these findings, we generated PDAC cell lines silenced for SHH and DISP1, the protein responsible for Hh ligand shedding. After confirming KD efficiency by RT-qPCR (Figure 1B), we investigated the effects of these perturbations on the expression of downstream signaling elements in co-cultured mouse fibroblasts. As expected, shSHH PANC-1 cells resulted in a significant reduction in Gli1 and Ptch1 expression levels in co-cultured fibroblasts (Figure 1C). DISP1 mediates the secretion of HH ligands and can be considered crucial for Hh crosstalk. Interestingly, DISP1 KD in PDAC cells caused an increase in both Ptch1 and Gli1 expression in the co-cultured MEFs, in line with our previous observations for ADAM metalloproteases [29].

Having generated these knockdown lines, we next aimed to determine the consequences of perturbed paracrine stromal Hh pathway activation in vivo. Subcutaneously grafting these KD lines, alongside a shCtrl condition, into mice revealed no changes in tumor formation/weight in either case (Figure 1D). This led us to conclude that altering Hh production and secretion in either direction does not influence PDAC tumor formation. To confirm in an orthogonal fashion that shSHH resulted in reduced stromal Hh signaling activity, we performed a co-culture experiment with Capan-2 cells as they exhibited enhanced trans-signaling capacity and were the only cells that exclusively expressed SHH, but not IHH or DHH [31]. As expected, despite being efficiently silenced for SHH, the Capan-2 cells exhibited no decrease in expression of downstream Hh pathway elements and retained a comparable GLI1 expression [31]. We next assessed the influence of shSHH Capan-2 on the Hh trans-signaling activity in co-cultured Shh LIGHT II reporter cells. Our experimental setup included shCtrl Capan-2 cells as well as an anti-SHH (5E1) blocking antibody (and an anti-Myc antibody (9E10) as a control) during co-culturing. We observed a strong reduction in Hh trans-signaling in the shSHH Capan-2 cells. This was comparable to the reduction induced by 5E1 inhibition, which reduced the signaling activity to levels observed in reporter cell monocultures (Figure 1E). Again, although shSHH KD led to a strong reduction in Hh trans-signaling activity, it did not affect tumor growth in vivo (Figure 1F).

Previous research has indicated an anti-correlation between GLI activity and SHH expression in PDAC cell lines as well as a correlation between the two and EMT score and PDAC tumor subtype [28]. Thus, we aimed to confirm whether our observation in Figure 1A still stands in a subtyped cohort of PDAC tumors, and with that, assessing the possibility of grouping patient samples based on ligand abundance. To that end, we compared the expression (Z-scores) of Hh ligands between the transcriptomic subtypes of PDAC using a publicly available bulk RNA-seq cohort. This revealed a lower expression of the ligands in the more aggressive squamous subtype (closely related to basal PDAC) (Figure 1G), which corroborates our earlier findings. Nonetheless, despite the above observations with PDAC transcriptomic subtype, we observed no consistent prognostic signal for either Hh ligand (SHH, IHH; Figure 1H) across PDAC gene expression datasets [32,33,34]. With this we concluded that Hh ligands alone are not sufficient to provide a strong prognostic signal and shifted our focus towards the stroma.

### 2.2. Hedgehog Response in the PDAC Microenvironment Impacts Tumor Progression

Moving on from the above findings, we interrogated the previously mentioned PDX RNA sequencing set and observed a link between epithelial ligand expression and stromal features (Appendix A). This reinforced our hypothesis that Hh effects are mediated through the stroma and prompted us to reassess the contribution of stromal Hh responses in vivo. To this end, we co-injected PDAC cells with either MEFs deficient or proficient for Smo. After confirming the presence of injected MEFs in the formed tumors (rather than host fibroblasts; Figure 2A), we evaluated the effect of stromal Smo expression on tumor formation. The presence of Smo-proficient mouse fibroblasts seemed to restrict tumor formation, as was evident by a marked reduction in tumor weights (Figure 2B). This confirmed the currently accepted notion that the impact of Hh signaling on PDAC growth is not effectuated by the tumor cells but mediated through the stroma.

To confirm the above notion, we next evaluated the influence of downstream Hh signaling elements (GLI1, PTCH1, and SMO), which we know to be stromal, on patient survival outcomes using independent PDAC cohorts. This, however, revealed no consistent prognostic signal (Figure 2C). Therefore, we reasoned that a PDAC-tailored signature would be more appropriate. To establish this, we turned to an in vitro co-culture with reciprocal tumor-stroma interactions. We utilized co-culture aggregates to determine stroma-derived transcripts that signal to the tumor compartment as previously described (Figure 2D) [31]. Having subjected the aggregates to transcriptome-wide quantitative sequencing, we mapped reads to either the human or mouse genome. This revealed a large percent of transcripts which were of human origin (Figure 2E). We next compared the expression of validated genes (as in known to be stromal or epithelial) to evaluate the accuracy of our species deconvolution. This confirmed that stromal transcripts (Fap and Acta2) were mapped to the mouse genome, while tumor cell-intrinsic transcripts (MUC1 and KRT19) were human reads (Figure 2F). To identify hits that are Hh-dependent and not simply those involved in stroma activation, co-cultures treated with an SHH-blocking antibody (5E1) or a control IgG, as previously described [31], and mouse specific reads were considered Hedgehog-responsive stromal genes (Figure 2D–F). We next aimed to apply these genes for the stratification of PDAC tumors based on stromal Hh response.

### 2.3. PDAC Tumors Characterized by High Stromal Hh Response Are Favorably Prognostic

To determine an informative set of genes reflective of stromal Hh response, we selected mouse genes that exhibited a 2-fold reduction, or more, in response to 5E1 treatment. Doing so resulted in a gene list, which we further filtered using gene ontology for extracellular proteins to identify potential reciprocal signals. This left us with 251 genes that were considered for further analysis (Appendix A; [31]). We then determined prognostic genes by Cox regression (*p* < 0.05) in the TCGA PAAD set and used those genes (both favorable and unfavorable) to stratify PDAC tumors based on their stromal Hh responsiveness (Appendix A; Figure 3A,B). We opted to cluster our discovery cohort into two groups, as our aim is comparing high vs. low responsiveness. Subsequently, to validate our findings, we used the identified genes to cluster two additional PDAC cohorts and observed similarly prognostic groups (Figure 3C). Our next step was comparing the expression of previously established stromal signatures between our clusters to better characterize Hh-responsive stroma. Moffitt et al. have described signatures that are indicative of stromal activation, i.e., activated and normal stroma, which also vary in terms of prognosis [30]. Similarly, Elyada et al. identified signatures that reflect different CAF phenotypes, namely myCAFs and iCAFs [3]. Comparing the expression of these signatures in the prognostic groups obtained after clustering the PAAD dataset revealed an enrichment for Moffitt normal stroma (Figure 3D) and Elyada iCAF (Figure 3E) signatures in the good prognosis group. We considered this an indication that upon activation, fibroblasts become aberrantly activated and no longer need Hh signals from surrounding PDAC cells. Furthermore, we noted that when tumors are low in the aforementioned stromal cues, they are more enriched for basal PDAC (Figure 3F), which would indicate that Hh responsive stroma can inhibit aggressive mesenchymal PDAC features.

## 3. Discussion and Conclusions

The Hh signaling pathway is a major driver of stromal content and CAF activation in PDAC [35]. For the most part, CAFs were assumed to be cancer-promoting, presenting as an ideal candidate for Hh-targeted therapy [4]. However, despite promising pre-clinical findings, Hh inhibition has not been successful in the clinic [20]. Heterogeneity of PDAC tumors, particularly within the stromal compartment, is thought to explain these poor clinical outcomes. The ability of PDAC cells to secrete Hh ligands is very well recorded in the literature, as well as the relation between ligand abundance and EMT features [27]. Despite this, we saw that Hh ligand expression by itself does not hold prognostic value. Instead, we observe that stromal cells are differentially responsive to Hh ligands, which may, in part, explain the poor performance of Hh-targeted modalities in the clinic. This prompted us to determine a signature to classify PDAC tumors based on Hh signaling.

Our work supports previous findings regarding the implications of Hh targeting in PDAC. We also confirmed that fibroblasts have altered Gli1 and Ptch1 expression in response to modifying Hh ligand secretion in PDAC cells. In fact, genetically perturbing Hh signaling in PDAC cells does not hamper their tumorigenic capacity; however, it does significantly reduce their trans-Hh signaling ability with surrounding fibroblasts. In addition, we challenged the idea that stroma is inherently bad and corroborated the findings of previous groups in this respect [24,36,37]. Indeed, co-injection of PDAC cells with Smo-proficient mouse fibroblasts seemed to restrict tumor formation compared to when Smo-proficient cells were used. Others have shown that Hh inhibitors can influence CAF composition and balance within a PDAC tumor. The Smo antagonist LDE225 has been shown to shift the CAF population towards an iCAF phenotype while simultaneously impairing tumor growth, suggesting that iCAFs are not inherently tumor-supportive as generally believed [24]. Our analyses here show a link between normal stroma and iCAFs with Hh ligands. This was confirmed by an in-depth comparison of clusters based on the identified Hh response genes. This revealed that favorable stromal features are those enriched for normal stroma and iCAF signatures, which would suggest that Hh pathway activation is not strictly associated with activated stroma or myofibroblast-like phenotypes as depicted in the literature. We also observed that tumors low in Hh-responsive stroma exhibit features of the aggressive basal subtype of PDAC. Thus, Hh-mediated crosstalk between PDAC cells and iCAFs may play a role in instilling PDAC phenotypes. It is possible that following their activation, stromal cells acquire an independence from Hh ligands. Considering the contentious role of CAF subsets on PDAC growth [25,36,38], we believe there is value in re-assessing the contributions of different CAF subsets on tumor growth. The presence of iCAFs appears to be a favorable scenario in PDAC, suggesting that the use of Hh inhibitors, such as LDE225, may be beneficial to at least a subset of PDAC patients, namely patients with basal-subtype tumors that are more prevalent in activated stromal features. Thus, appropriate stratification tools may be key to improving patient outcomes in response to Hh-targeted therapies.

Developments in the past decade have resulted in approaches to target the stroma in PDAC and reduce its characteristic desmoplastic reaction. The phase III HALO-301 trial evaluated the use of pegylated recombinant human hyaluronidase in combination with gemcitabine with nab-paclitaxel with the purpose of reducing the interstitial pressure of PDAC tumors and enhancing drug delivery (NCT02715804, [39]). Unfortunately, however, this combination did not result in any improvements in patient survival. Other clinical trials instead aimed to inhibit stroma-promoting pathways in PDAC such as TGF-β (NCT03563248), FAK (NCT02546531), IL1R (NCT02550327), or CTGF (NCT03941093). These trials revealed either no added benefit of the stroma-targeting agents or have not yet published their findings. Another approach sought to impede the activation of pancreatic stellate cells, as they are precursor cells that contribute to the CAF pool. Vitamin D receptor activation showed great promise in this regard [40] and resulted in clinical trials that are still ongoing (NCT04054362; NCT04524702). Trans Retinoic Acid (ATRA) also sparked great interest and is currently being investigated in a phase II clinical trial (STARPAC2; NCT04241276) in combination with standard chemotherapy (gemcitabine with nab-paclitaxel) for the treatment of patients with locally advanced, inoperable, non-metastatic PDAC [41]. Despite this number of stroma-targeted therapies and combinatorial regimens, we have yet to see a significant impact on the clinic. We should re-assess our approach to stroma targeting for PDAC. Considering our findings here and indications in the literature, we believe that heterogeneity within the stromal compartment should be a consideration before implementing these approaches.

In this work, we raise patient stratification as a potential strategy for improving the clinical performance of Hh-targeted treatment modalities (Figure 4). This may allow for determining the best course of treatment, specifically for combinatorial therapies comprising Hh-targeted therapeutic regimens. We propose that future clinical trials focus more on patient stratification and stromal phenotypes prior to administration of Hh antagonists.

## 4. Limitations and Perspectives

This study is not free of limitations. While we successfully identified a signature that can be used for PDAC patient stratification, this signature was derived from in vitro co-cultures comprising murine fibroblasts. This may be a source of bias and might limit the clinical significance of our findings. To circumvent this and confirm what we found, future studies should compare whether good and bad responders to Hh-targeted therapy in controlled settings (clinical trials) relate to our identified groups. This may reveal if our stratification method has potential in guiding the use of Hh-targeted modalities in the clinic. Moreover, further characterization as to why hedgehog-responsive stroma results in a more favorable prognosis and whether this is mediated through cellular processes such as autophagy and/or involves the immune system should be further explored. Nonetheless, we believe the findings presented herein offer some insights to the field and emphasize the value of patient stratification.

## 5. Materials and Methods

### 5.1. Cell Culture and Generation of Antibodies

Cell lines were maintained in Dulbecco’s Modified Eagle Medium (DMEM, Lonza, Basel, Switzerland) containing 8% FBS, l-glutamine (2 mm), penicillin (100 units/mL), and streptomycin (500 μg/mL) and cultured in adherence to routine cell culture practices (37 °C; 5% CO_2_; 95% humidity). Shh-LIGHT II reporter cells were grown under the abovementioned conditions both with additional supplementation with neomycin (400 μg/mL) and zeocin (150 μg/mL).

The following ATCC lines were utilized in this work: PANC-1 (RRID:CVCL_0480), Capan-2 (RRID:CVCL_0026), HEK293T (RRID:CVCL_0063), and C3H10T1/2 (RRID:CVCL_0190). Moreover, 9E10 and 5E1 hybridoma cells (Developmental Studies Hybridoma Bank, Iowa City, IA, USA) were maintained in RPMI containing 10% FBS, L-glutamine, penicillin, and streptomycin (all from Lonza). Hybridoma cells were switched to serum-free medium for generation of 9E10 and 5E1 antibodies. Cells were monitored for mycoplasma monthly and STR profiled (human cell lines) no more than 6 months prior to performing experiments.

### 5.2. Co-Culture Model

A 3-dimensional co-culture system was utilized in this work to best model the tumor–stromal interactions. In short, cancer cells and fibroblasts (1:1 ratio) were cultured on non-adherent dishes (60 × 15 mm; Greiner, Kremsmünster, Austria) in DMEM supplemented with 0.5% FCS and were subjected to shaking (rotary shaker at 55 rpm). These conditions resulted in the formation of mixed aggregates, which were cultured for 5 d prior to further analysis.

### 5.3. Gene Silencing

SHH and DISP1 silencing of PDAC cell lines was achieved by lentiviral delivery of short hairpin RNA (shRNA). Lentivirus was produced by transfecting HEK293T cells (RRID:CVCL_0063) with Mission TRC library pLKO.1 transfer as described in the following paragraph. Subsequently, the virus-containing supernatant was harvested (at 48 h and 72 h), filtered through 0.45 μm filters (Millipore, Billerica, MA, USA), and stored at −80 °C until use.

Based on performance in our previous work and appropriate KD efficiency, the following shRNA were utilized herein: CTACGAGTCCAAGGCACATAT (TRCN0000033304) and CATTTCAGAAGCATCTCGAA for perturbing SHH and DISP1, respectively. The scrambled shc002 (RRID:Addgene_1864) was used as a control.

### 5.4. Lentivirus Generation and Transduction

Lentivirus was produced by transfecting HEK293T cells (RRID:CVCL_0063) with Mission TRC library pLKO.1 plasmid (RRID:Addgene_10878) containing the appropriate TRC clone or sequence (mentioned above) together with the envelope/packaging plasmids pMD2.G (RRID:Addgene_12259), pMDL (RRID:Addgene_12251), and pRSV-REV (RRID:Addgene_12253) using Lipofectamine 2000 transfection reagent (Thermo Fisher, Waltham, MA, USA). Subsequently, virus-containing supernatant was harvested (at 48 h and 72 h), filtered through 0.45 μm filters (Millipore, Billerica, MA, USA), and used for transduction. Cell lines at 70% confluency were transduced with concentrated lentivirus (1:2 ratio) and 8 μg/mL polybrene (Sigma, St. Louis, MO, USA) overnight. Two days after transduction, the cells were selected and maintained in a culture medium containing 2 μg/mL puromycin (Invitrogen, 11668019) for at least two passages or sorted by fluorescent signal before subsequent analyses were performed. Target transcript levels were determined by qRT-PCR.

### 5.5. Quantitative RT-PCR

RNA isolation was performed (Macherey Nagel, Düren, Germany), and cDNA was synthesized using Superscript III (Invitrogen). Quantitative PCR (qRT-PCR) was performed using SYBR green reagent and the Lightcycler LC480II system (both from Roche, Basel, Switzerland) following the manufacturer’s guidelines. The comparative threshold cycle (Cp) method was used to calculate transcript levels, which were normalized to a housekeeping gene (GAPDH/Gapdh), and data were represented as percent of respective scrambled control. The primer pairs used were the following:hGAPDH Fw: GAAGGTGAAGGTCGGAGTC; Rv: TGGAAGATGGTGATGGGATThSHH Fw: GCTCGGTGAAAGCAGAGAAC; Rv: CCAGGAAAGTGAGGAAGTCGhDISP1 Fw: ACAGCTTTTTCTGCGACGTT; Rv: TCCATAATGTCTCCCCTCCAmGapdh Fw: CTCATGACCACAGTCCATGC; Rv: CACATTGGGGGTAGGAACACmGli1 Fw: ATAGGGTCTCGGGGTCTCA; Rv: CGGCTGACTGTGTAAGCAGAmPtch1 Fw: CTCAGGCAATACGAAGCACA; Rv: GACAAGGAGCCAGAGTCCAG

### 5.6. Preparation of RNA-Seq Libraries

RNA was extracted from co-cultures (three separate experiments) treated with 5E1. Treatment efficiency was confirmed by RT-PCR for mPtch1, and libraries were constructed using the mRNA-Seq Sample Preparation Kit (Illumina, Hayward, CA, USA) following the manufacturer’s instructions. Sequencing was performed using an Illumina 1 G Genome Analyzer. Reads were non-paired end, 31 bp, and mapped and annotated to both human and mouse genomes using ArrayStar 3.0 software (DNASTar, Madison, WI, USA) with the Qseq module. Human orthologs of mouse genes were downloaded from MGI (http://www.informatics.jax.org/), and Gene Ontology analysis using the GO cellular component term “extracellular region” was performed.

### 5.7. Animals and PDAC PDX Model

Immunocompromised NSG mice (JAX 005557) were utilized in this study. The animals were bred and maintained at the local animal facility according to the legislation and after ethical approval (DIX102511 and DTB102306). In short, control PDAC cells, or those KD for SHH/DISP1, were injected into the flanks of the mice, and tumors were harvested at the same timepoints for an appropriate tumor size comparison. Co-injections between PDAC cells and Smo-/- MEFs transduced with either SmoWT or an empty vector (as previously described [42]) were also performed, and tumors were harvested when control tumors reached a size of around 1000 mm^3^.

### 5.8. Reporter Assay

Shh-LIGHT II stably transfected reporter fibroblasts were co-cultured with PDAC cells for 5 days, after which they were lysed with passive lysis buffer following Promega’s guidelines. Luciferase activity was then measured using the Promega Dual-Glo Luciferase Assay System (Promega) on a Victor plate reader (PerkinElmer, Waltham, MA, USA). Each Firefly luciferase value was corrected for its CMV-driven Renilla luciferase standard to negate nonspecific effects. 5E1 and 9E10 antibodies were incorporated in this assay as positive and negative controls for HH inhibition, respectively.

### 5.9. Slide Preparation and Immunohistochemical Staining

Tumors were fixed in 4% formalin overnight prior to paraffin embedding. Tissue sections (5 μM) were deparaffinized, after which antigen retrieval was performed by adding 10 mM of sodium citrate solution and boiling (15 min). Endogenous peroxidase activity was blocked with 3% hydrogen peroxide in PBS. Non-specific staining was blocked using 5% normal goat serum for 20 min at room temperature. Primary antibodies were diluted in normal antibody diluent (KliniPath, Duiven, The Netherlands) applied on tissue sections and incubated overnight at 4 °C in a humidified chamber. Subsequently, sections were washed with PBS and then incubated with BrightVision poly HRP-anti IgG (Immunologic, Duiven, The Netherlands) for 30 min at RT. Staining visualization was performed with vector novaRED peroxidase (Vector) according to manufacturer’s protocol. This was followed by counterstaining with 30% haematoxylin and mounting using non-aqueous medium. Antibodies used for immunohistochemistry were as follows: anti-CK19 (MU246-UC, BioGenex, Fremont, CA, USA); anti-Myc (SC-040, Santa Cruz, Dallas, TX, USA); anti-αSMA (ab5694, abcam, Cambridge, UK). All were used at 1:200 dilution. Stained PDX slides were photographed on an Olympus BX50 microscope with camera (Olympus, Tokyo, Japan).

### 5.10. Statistics and Bioinformatic Analyses

Statistical analyses were performed using the R2 Genomics Analysis and Visualization Platform (RRID:SCR_025770; https://r2.amc.nl), R (R Project for Statistical Computing; RRID:SCR_001905), and GraphPad Prism 5.02 (RRID:SCR_002798). In short, continuous variables were compared between groups using an unpaired *t*-test, One Way Analysis of variance (ANOVA), or Mann–Whitney U test. The Pearson Chi-square test and the two-sided Fisher’s exact test were utilized to compare the distribution of categorical variables. Kaplan–Meier curves were used to analyze differences in survival, and the log-rank test was used to calculate *p*-values. Correlation coefficients and significance were calculated. Significant values were indicated on the graphs as well as in the figure legends either as exact values or as follows: *, *p* ≤ 0.05; **, *p* ≤ 0 0.01; ***, *p* < 0.001; ****, *p* < 0.0001.

## Figures and Tables

**Figure 1 ijms-26-05200-f001:**
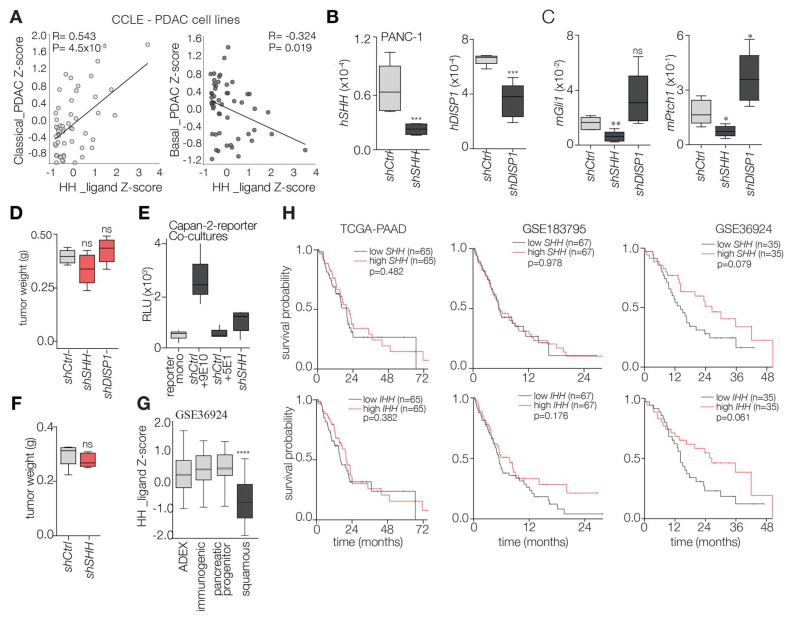
**Hh ligand abundance is not prognostic in PDAC**. (**A**) Association between Hh ligand and Moffitt subtype gene signatures [30] (Z-scores) was assessed in PDAC cell lines annotated in the CCLE dataset. Correlation coefficients and significance were calculated and included on the graphs. (**B**) PANC1 cells were silenced using shRNAs against SHH and DISP1, and qRT-PCR was performed for the transcripts shown. Data are presented as box plots showing mean ± SD of three replicates, and statistical significance was assessed by unpaired *t*-test. ***, *p* < 0.001; ns, not significant; *p* < 0.0001. (**C**) PANC1 cells silenced for SHH and DISP1 were co-cultured with mouse fibroblasts after which qRT-PCR was performed for the shown mouse transcripts. Data are presented as box plots showing mean ± SD of three replicates, and statistical significance was assessed by unpaired *t*-test. *, *p* < 0.05; **, *p* < 0.01. (**D**) Silenced, or control, PANC1 cells were injected in mice to determine effects on tumor formation. After the experimental endpoint, tumors were harvested and weighed. Data are presented as box plots showing mean ± SD of five tumors for each group, and statistical significance was assessed by unpaired *t*-test. (**E**) Capan-2 cells were silenced for SHH (as in (**B**)) and co-cultured with Shh LIGHT II reporter cells to assess Hh trans-signaling activity. The experimental setup includes mono-cultured reporter cells (light grey) as well as co-cultures of shCtrl Capan-2 cells (dark grey) with a positive, anti-SHH (5E1) blocking antibody, and a negative (anti-Myc antibody; 9E10) control. (**F**) same as in (**D**) but using Capan-2 cells. (**G**) The expression of Hh ligands (Zscore) was assessed in the indicated primary PDAC set with associated subtype labels (GSE36924). Statistical significance was determined by comparing expression between all other subtypes and squamous by unpaired *t*-test; ****, *p* < 0.0001. (**H**) Kaplan-Meier survival analysis for PDAC patients in three publicly available cohorts (TCGA-PAAD; GSE183795; GSE36924) based on median expression of Hh ligands (SHH; IHH). *p*-value was determined by log-rank and is presented in the graphs.

**Figure 2 ijms-26-05200-f002:**
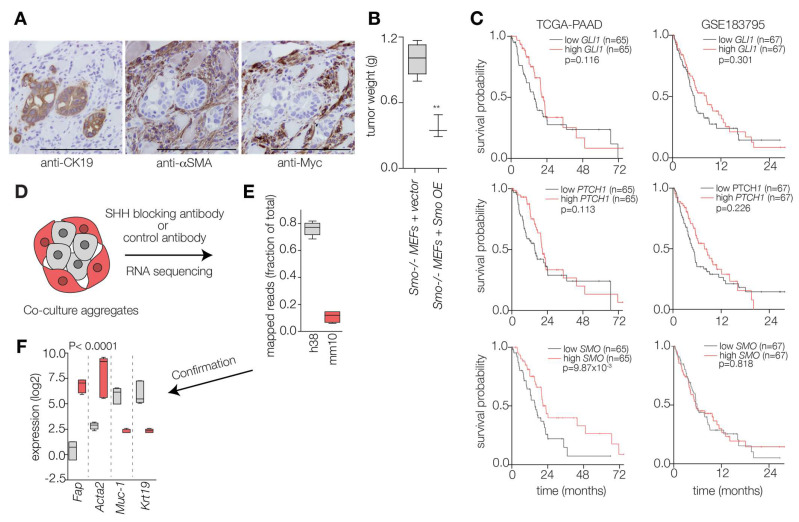
**The effects of PDAC-derived Hh ligands are mediated through the stroma**. (**A**) PDAC cells were co-injected with MEFs (Smo-/-) transduced with an empty vector or wt Smo. Subsequently, IHC was performed on the emergent PDXs to confirm the presence of injected MEFs in the formed tumors (rather than host fibroblasts). Scale bar indicates 200 µm. CK19 was used as an epithelial stain, whereas αSMA and Myc were included as stromal markers. (**B**) Formed PDXs in the two groups were then assessed to evaluate the impact of stromal Smo on tumor growth in vivo. Data are presented as box plots showing mean ± SD of at least three tumors per group, and statistical significance was assessed by unpaired *t*-test. **, *p* < 0.01. (**C**) Kaplan–Meier survival analysis for PDAC patients in publicly available cohorts (TCGA-PAAD; GSE183795) based on median expression of Hh signaling elements (GLI1; PTCH1; SMO). *p*-value was determined by log-rank and is presented on the graphs. (**D**) Co-culture aggregates comprising MEFs and PDAC cells, in the presence of an SHH blocking antibody (5E1) or a control antibody, were established, and RNA was isolated. (**E**) Samples were then subjected to transcriptome-wide quantitative sequencing. Obtained reads were aligned to either the mouse (red) or human (grey) genome to allow differentiation between stroma-derived and tumor cell-derived transcripts. Unique fractions of mapped reads are presented as box plots. (**F**) Prior to further analysis, transcripts of a validated origin (stromal in grey or epithelial in red) were compared between the species to confirm the pipeline accuracy. Data are presented as box plots showing mean ± SD of three replicates, and statistical significance was assessed by two-way ANOVA.

**Figure 3 ijms-26-05200-f003:**
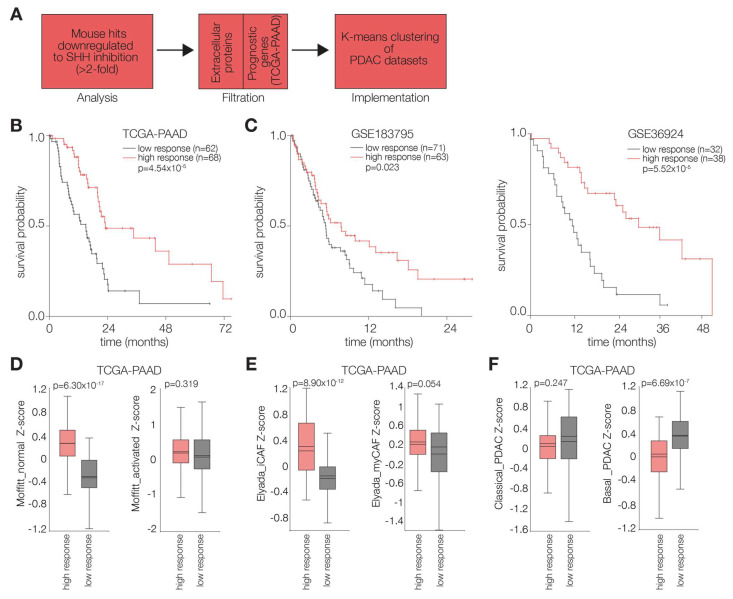
**The effects of PDAC-derived Hh ligands are mediated through the stroma**. (**A**) The analysis workflow following read mapping. In short, reads mapped to the mouse genome were processed and filtered to ultimately determine a set of genes which we used for clustering PDAC cohorts. (**B**) Kaplan–Meier survival analysis on the TCGA-PAAD (discovery) cohort after K-means clustering using a nine-gene stromal response signature. *p*-value was determined by log-rank and is presented on the graphs. (**C**) Validation of the aforementioned signature in two independent publicly available cohorts (GSE183795; GSE36924). *p*-value was determined by log-rank and is presented on the graphs. The expression of gene signatures (Zscore) for Moffit stromal subtypes [30] (**D**), Elyada CAF subtypes [3] (**E**), and Moffit PDAC subtypes [30] (**F**) were assessed in the groups obtained after clustering the TCGA-PAAD cohort (as shown in (**B**)). Statistical significance was determined by unpaired *t*-test, and *p* values are indicated on the graphs.

**Figure 4 ijms-26-05200-f004:**
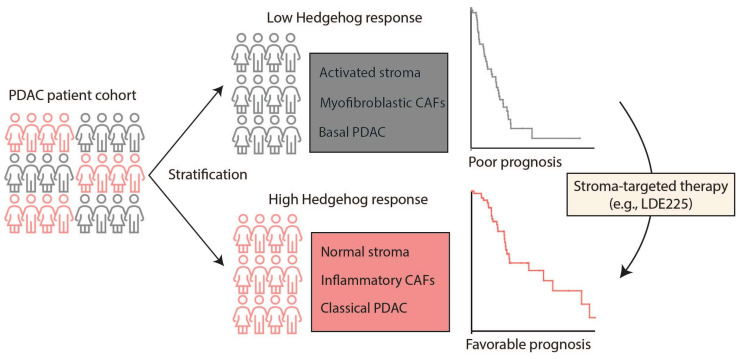
Diagram depicting how patient stratification can guide and improve PDAC clinical care.

## Data Availability

Data are available from the corresponding author following reasonable request.

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
