# Peer review of "Stromal Hedgehog Signaling Is Associated with Favorable Outcomes in Pancreatic Cancer"

_ijms, 2025, doi:10.3390/ijms26115200_

Round 1
Reviewer 1 Report
Comments and Suggestions for Authors
In this manuscript, Manoukian et al have investigated the role of Hedgehog (Hh) pathway and the role of stroma in PDAC progression. Through their in vitro and in vivo experiments, the authors highlight that Hh ligands alone do not influence PDAC tumor progression. The PDAC tumors that exhibit high stromal Hh signals have a prognostic value, and this can be used to design the treatment strategy to better patient outcomes.
Mentioned below are my comments:
Major comments
1) Figure 3D-F: The error bars between the high response group and the low response group are overlapping. Please explain the large error bars.
2) A diagram depicting patient stratification based on stromal markers for PDAC treatment in the discussion section will facilitate capturing key points explained in the discussion section.
Minor comments
1) Few words are written twice. Please correct the typographical errors in the following sentences:
a) Abstract : line 31 . Work “ to “ is repeated.
b) Line 50, “that’’ word is repeated.
c)Line 112, “ GLI1” is repeated
d) Line 189, “us’’ word is repeated
e) Caption 2c is repeated line 226
2)Figure 3 caption is mentioned as Figure 2.
Author Response
Reviewer 1:
We appreciate the constructive suggestions of the reviewer and have done our best to address your concerns. Please note a point-by-point reply to your comments.
Major comments
Comment: 1) Figure 3D-F: The error bars between the high response group and the low response group are overlapping. Please explain the large error bars.
Reply: We believe the observed error bars are a normal phenomenon reflecting the inter-patient heterogeneity of PDAC and is expected considering this is bulk RNA-sequencing data. For transparency, we have provided a version of the plots with data points.
2) A diagram depicting patient stratification based on stromal markers for PDAC treatment in the discussion section will facilitate capturing key points explained in the discussion section.
Reply: As per the reviewer’s recommendation, we have added a diagram to showcase how patient stratification can inform the clinic (Fig 4).
Minor comments
1) Few words are written twice. Please correct the typographical errors in the following sentences:
- a) Abstract : line 31 . Work “ to “ is repeated.
- b) Line 50, “that’’ word is repeated.
c)Line 112, “ GLI1” is repeated
- d) Line 189, “us’’ word is repeated
- e) Caption 2c is repeated line 226
2)Figure 3 caption is mentioned as Figure 2.
Reply: we have addressed these errors and some others we spotted after closer scrutiny.

Reviewer 2 Report
Comments and Suggestions for Authors
The current manuscript is interesting and may be considered for publication after significant revisions.
1. It is recommended that the authors incorporate a thorough discussion of ongoing research on signaling linked to positive outcomes in pancreatic cancer.
2. The introduction should be revised to include the paragraph relating to point 1, in order to cover a broad and comprehensive explanation of current work.
3. Add a new section addressing the strengths, limitations, and perspectives in the revised version if possible.
4. Are there any insights into how this work is related to autophages?
Author Response
Reviewer 2:
The co-authors appreciate your feedback and have done our best to incorporate your suggestions into the manuscript. Please note a point-by-point reply to your comments.
1.It is recommended that the authors incorporate a thorough discussion of ongoing research on signaling linked to positive outcomes in pancreatic cancer.
Reply: We have added some sentences relating to important signaling pathways in PDAC in the introduction (lines 59-73) and presented current therapeutic regimens and approaches that are being evaluated in clinical trials in the discussion section (334-354).
- The introduction should be revised to include the paragraph relating to point 1, in order to cover a broad and comprehensive explanation of current work.
Reply: We have further elaborated on relevant signaling pathways in PDAC, especially those positively prognostic, to provide the reader with a good overview and highlight why we are looking more in-depth into the stroma. Kindly refer to lines 58-71.
- Add a new section addressing the strengths, limitations, and perspectives in the revised version if possible.
Reply: An appropriate section has been added as per your suggestion titles “limitations and perspectives”.
- Are there any insights into how this work is related to autophages?
Reply: Although we have not explored autophagy in the current work, we believe this may very well be a contributing factor. The link between the Hedgehog pathway and autophagy has been previously described in literature (PMID: 23149744). There is also evidence linking this pathway with CAF formation, metabolic reprogramming and tumor-stroma crosstalk (PMID: 3823468). We performed some analyses and saw that Hh-responsive stroma group was is more enriched for a GO signature for autophagy (see panel A below). We also saw that iCAFs had a higher expression compared to quiescent or activated (my)CAFs using data from the initial paper that described the iCAF subset (B). We opted to not add this to the paper as it is not our main message but could include it at the discretion of the editor. Thank you for pointing this out.
Figure R1. The association between autophagy and stromal-response to Hh signaling. (a) the expression (Zscore) of a gene signature for the positive regulation of autophagy (GO:0010508) was assessed in the groups obtained after clustering the TCGA-PAAD cohort (as shown in Fig 3b). (b) similarly, the same signature was projected on a publicly available dataset (GSE93313) comparing the influence of culture conditions on pancreatic stellate cells. In short, 3D co-cultures with cancer cells, 3D mono-cultures, and 2D mono-cultures are compared and represent iCAFs, quiescent PSCs, and myCAFs, respectively. Statistical significance was determined by unpaired T test or ANOVA and P values are indicated on the graphs.
